# Three Myocardial Diseases in One Heart: Arrhythmogenic Right Ventricular Cardiomyopathy, Left Ventricular Noncompaction and Myocarditis

Yulia Lutokhina [1,*], Olga Blagova [1], Nadezhda Varionchik [1], Svetlana Alexandrova [2], Nina Gagarina [1], Eugenia Kogan [1], Vsevolod Sedov [1], Anna Shestak [3], Elena Zaklyazminskaya [3] and Alexander Nedostup [1]

[1] Department of Faculty Therapy, I.M. Sechenov First Moscow State Medical University (Sechenov University), 119991 Moscow, Russia; blagovao@mail.ru (O.B.); vanadya@gmail.com (N.V.); nina-gagarina-ct@yandex.ru (N.G.); koganevg@gmail.com (E.K.); vps52@mail.ru (V.S.); avnedostup@mail.ru (A.N.)

[2] Radiology Department, A.N. Bakoulev Center for Cardiovascular Surgery RAMS, 121552 Moscow, Russia; svaleksandrova@yandex.ru

[3] Laboratory of Medical Genetics, B.V. Petrovsky Russian Research Center of Surgery, 119991 Moscow, Russia; anna.shestak87@gmail.com (A.S.); helenezak@gmail.com (E.Z.)

* Correspondence: lebedeva12@gmail.com; Tel.: +7-(917)-5963907

**Abstract:** Purpose: To evaluate the clinical features, laboratory and instrumental tests results and the effectiveness of complex treatment in a patient with multiple etiologies of dilated cardiomyopathy (DCM) with a high risk of sudden cardiac death. Methods: Female patient was 34 years old. Follow up period was seven years. Since the age of 23 (after a respiratory infection), chest pains and shortness of breath appeared. Coronary arteries were intact. After syncope in 2013, Holter-ECG was performed: 2048 premature ventricular beats (PVBs)/day and episode of sustained ventricular tachycardia (VT, 1 min) were registered. MRI was performed, and a cardioverter defibrillator (ICD) was implanted. Results: ECG showed low QRS voltage and negative T waves in leads V2-V6, III, aVF. In signal-averaged ECG, late potentials were detected. Echocardiography (EchoCG) demonstrated left and right ventricular dilatation, diffuse reduction of left ventricular (LV) contractility and multiple pseudochordae in LV. MRI showed LV noncompaction (LVNC), thickening of the epicardial fat and hypo-/dyskinesia of the anterior wall of the right ventricular (RV), dilatation of both ventricles with decrease of their ejection fraction and subepicardial gadolinium enhancement in the early and late phase in the LV, intraventricular septum and the free walls of the RV. The presence of LVNC was confirmed by cardiac computed tomography (CT). Late contrast enhancement in the middle and subendocardial layer of the LV was observed as well. The level of anticardiac antibodies was high (1:160–1:320). The reasons for statement of a possible diagnosis of myocarditis in this case were the connection of the onset of symptoms with viral infection, high titers of anticardiac antibodies, and early and late subepicardial contrast enhancement by MRI and CT. The endomyocardial biopsy was obtained, and subendocardial lipomatosis, separation of myocardium by fibrous septa, lymphocytic infiltrates (more than 14 cells/mm$^2$) and vasculitis were found. Viral genome in myocardium was not detected. A new splicing mutation in the desmoplakin (*DSP*) gene was found (NM_004415.4: c.1141-2A>G/N (rs794728111)). Combination of arrhythmogenic right ventricular cardiomyopathy (ARVC), LVNC and myocarditis was diagnosed. Immunosuppressive therapy (prednisone and azathioprine) was prescribed, LV ejection fraction stabilized at the level of 40%. The appropriate shocks of the ICD due to sustainedVT (HR 210/min) with transformation into ventricular fibrillation were recorded twice. For this reason, sotalol was temporarily replaced with amiodarone. After the suppression of myocarditis activity, sustained VT and ICD interventions were not observed. Conclusions: In a young patient with arrhythmogenic syncope and DCM syndrome, a combination of ARVC (two major and three minor criteria, definite diagnosis) and LVNC with the biopsy proved virus-negative chronic myocarditis was diagnosed. DCM as a syndrome can have multiple causes, and the combination of myocarditis and primary cardiomyopathy is not rare. LVNC can be observed in patients with typical desmosomal protein mutations. The use of immunosuppressive therapy led to the stabilization of heart failure and decreased the risk of arrhythmic events.

**Keywords:** arrhythmogenic right ventricular cardiomyopathy/dysplasia; left ventricular noncompaction; myocarditis; premature ventricular beats; ventricular tachycardia; endomyocardial biopsy

## 1. Introduction

The problem of diagnostics and treatment of myocardial diseases in the practice of modern cardiology is of great importance. Even more of an actual issue, is the diagnosis of a combination of several myocardial diseases. This paper will discuss the combination of three nosological entities simultaneously: arrhythmogenic right ventricular cardiomyopathy (ARVC), left ventricular noncompaction (LVNC) and myocarditis.

ARVC is an inherited myocardial disease characterized by right ventricle (RV) fibrofatty replacement, ventricular arrhythmias and a high risk of sudden cardiac death (SCD) [1]. ARVC was described in 1977 and, at that time, was considered to be a rare disease; however, later, as a result of improvement of diagnostic criteria [2], including the development of MRI criteria, information about its prevalence has changed. At present, the incidence of this cardiomyopathy varies depending on the population, from 1:1000 to 1:5000 [3,4].

LVNC is characterized by intensively developed ventricular trabeculae combined with deep intertrabecular lacunas lined with endocardium, not connected with coronary blood flow. It is generally accepted that LVNC syndrome in adults was first described by Chin et al. in 1990 [5], but LVNC was included in cardiomyopathy classification only in 2008. [6]. The incidence of LVNC, according to different data, varies from 0.014% to 1.3% [7]. The diagnosis of this cardiomyopathy is based on three imaging techniques: echocardiography (EchoCG), cardiac magnetic resonance imaging (MRI) and cardiac multispiral computed tomography (CT). Diagnostic criteria have been developed for each of these methods [5,8–10].

Myocarditis is defined as an inflammatory myocardial disease, diagnosed on the basis of established histological, immunological and immunohistochemical criteria [11]. It is difficult to estimate its prevalence in the population, since endomyocardial biopsy (EMB), the "gold" standard of in vivo diagnosis of myocarditis, is performed infrequently. The incidence of morphologically verified myocarditis among young patients who died suddenly reaches 42%, and among adults and children with dilated cardiomyopathy (DCM), myocarditis in EMB is detected in 16% and 46% of patients, respectively [12]. Primary genetically determined cardiomyopathy was thought to be a favorable background for superimposed myocarditis; moreover, the inflammatory component could contribute to realization of the abnormal genetic program [13,14].

## 2. Case Presentation

Patient I, 34 years old, was admitted to the Vinogradov Faculty Therapeutic Clinic (FTC; Sechenov University) on 2 December 2013. She complained of palpitations, dyspnea provoked by ordinary physical activity, recurrent leg edema and general weakness.

### 2.1. Social and Family History

The patient is a philologist and journalist. She does not smoke or abuse alcohol. There is no family history of cardiomyopathies; her mother (64 years old) has rheumatoid arthritis, Sjögren's syndrome and autoimmune pancreatitis.

### 2.2. Medical History

The patient had thrombocytopenic purpura at the age of two, she was treated with prednisolone and complete remission was achieved. At the age of 18, she was diagnosed with chronic autoimmune thyroiditis, hypothyroidism, for which she receives hormone replacement therapy (L-thyroxine).

Patient noticed onset of chest pain in 2000 (20 years old), she did not undergo any investigation and did not receive any treatment (Figure 1). In 2003, after an upper respiratory tract infection with a prolonged (up to one month) period of subfebrile fever, chest pain became more intensive and dyspnea appeared. EchoCG in 2005 revealed mitral valve prolapse with first degree regurgitation and left ventricular (LV) dilatation, with decreased ejection fraction (EF) to 42%. After these findings, she underwent EchoCG annually and the cardiac contractility remained on the same level.

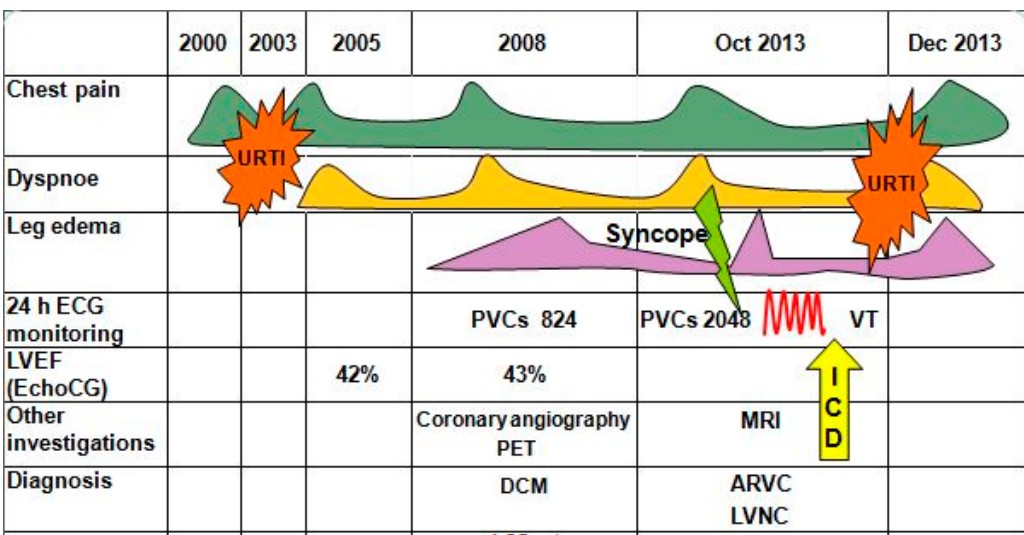

**Figure 1.** Scheme of the medical history (explanation in the text). ARVC—arrhythmogenic right ventricular cardiomyopathy; DCM—dilated cardiomyopathy; ECG—electrocardiography; EchoCG—echocardiography; ICD—implantation of cardioverter-defibrillator; LVEF—left ventricular ejection fraction; LVNC—left ventricular noncompaction; MRI—magnetic resonance imaging; PET—positron emission tomography; PVCs—premature ventricular contractions; URTI—upper respiratory tract infection.

The deterioration of her condition started in September 2007. On the back of psycho-emotional stress and fatigue, she was experiencing constant stabbing pain in the left side of her chest, which was self-limited, and increased dyspnea. In April 2008, she was hospitalized at the A.N.Bakoulev Center for Cardiovascular Surgery. EF remained on the same level (43%), and myocardial radionuclide perfusion scan revealed decreased accumulation in the area of interventricular septum (IVS). During 24 h ECG monitoring, 865 premature ventricular beats (PVBs) of three morphologies were registered. During coronary angiogram, no hemodynamically significant stenoses were found. Myocardial positron emission tomography was performed, and moderate decrease in perfusion and glucose metabolism in the middle sections of the IVS was registered, which was most likely caused by local thinning or local myocardial fibrosis. Diagnosis was formulated as dilated cardiomyopathy (DCM). Patient was prescribed perindopril 8 mg/day, bisoprolol 1.25 mg/day, trimetazidine 70 mg/day, furosemide 40 mg/day, spironolactone 25 mg/day and L-thyroxine 50 mg/day. Her condition remained stable. EchoCG performed in 2009, 2012 and 2013 showed no significant dynamics. EF varied from 38% to 40%.

The next deterioration of well-being was in the summer of 2013, when pre-syncope appeared. In September 2013, Holter monitoring recorded 2048 PVBs of three morphologies and an episode of sustained ventricular tachycardia (VT) lasting 1 min, accompanied by loss of consciousness. She was hospitalized at the A.N.Bakoulev Center for Cardiovascular Surgery, where she underwent cardiac MRI. The MRI revealed the epicardial fat thickening along the anterior wall of RV and posterior wall of LV with signs of "crawling" on myocardium, RV (end-diastolic diameter (EDD) 48 mm) and LV (EDD 66 mm) dilatation with decrease of EF of both ventricles (RV EF 25% and LV EF 41%), dilatation of LV outflow

tract (20 mm), edema of LV and prominent fibrous changes of non-ischemic genesis of RV and LV. The condition was thought to be chronic myocarditis, but ARVC and LVNC were not ruled out either. On 30 October 2013, a cardioverter-defibrillator (ICD) was implanted. After upper respiratory tract infection in November 2013, the patient noticed increase of chest pain, dyspnea and leg edema. She was admitted to the Department of Cardiology, FTC, for further examination and choice of treatment strategy.

### 2.3. Physical Examination

Heart sounds were regular, clear, heart rate 60 per min, blood preassure 90/60 mm Hg. Lungs auscultation revealed vesicular breathing, no rales, respiratory rate 20/min. Liver and spleen were not enlarged. Moderate legs edema was observed.

### 2.4. Laboratory Tests

Full blood count, biochemical panel, coagulogram and urinalysis showed no abnormalities. Anticardiac antibodies (Ab) test was performed to verify possible chronic myocarditis, showing specific antinuclear factor to cardiomyocyte nuclei (ANF) 1:320 (normally negative), Ab to endothelial antigens (AbE) 1: 160 (N ≤ 1:40), Ab to cardiomyocyte antigens (AbC) 1:160 (N ≤ 1:40), Ab to smooth muscle antigens (AbSM) 1:160 (N ≤ 1:40), and Ab to cardiac conductive fibers antigens (AbCF) 1:320 (N ≤ 1:40). No genome of cardiotropic viruses (herpetic group viruses, parvovirus B19 and cytomegalovirus) were detected in the blood.

### 2.5. Instrumental Tests

*ECG* on admission (Figure 2) showed low QRS voltages in the limb leads, and negative T waves in the left precordials (minor ARVC criterion) and in the inferior leads (indicating LV involvement). The signal-averaged *ECG* revealed late potentials (minor ARVC criterion): filtered QRS = 137 ms (N < 114 ms), low-amplitude signal duration = 40 ms (N < 38 ms) and root-mean-square voltage of terminal 40 ms = 26 μV (N > 20 μV). On 24 h *ECG* monitoring (treatment: sotalol 160 mg/day) 1700 PVBs (minor ARCV criterion), 54 couplets and one triplet were registered.

EchoCG showed left and right ventricular dilatation (EDD 6.2 and 4.0 cm, respectively) and diffuse reduction of LV contractility (EF 37%), with normal VTI 15.4 cm. Multiple pseudochordae were visualized in LV, but there were no convincing echocardiographic signs of LVNC. To verify the presence of LVNC, the patient underwent cardiac computed tomography (CT); coronary arteries were intact, LV myocardium had increased trabecularity in the apical-lateral and posterior walls, and the ratio of noncompact and compact layers was 3:1. Zones of late contrast agent accumulation in the middle and subendocardial layer of LV were visualized. In addition, a myocardial radionuclide perfusion scan was performed, which showed indicator inclusions in LV myocardium with diffusely nonuniform distribution, with areas of relative indicator hypoaccumulation more pronounced in the IVS and apical and middle sections of LV anterior wall. Such pattern is nonspecific, but typical for myocardial diseases. Cardiac MRI CD-disk was reanalyzed by Professor Sinitsyn. MRI findings were not typical for postinflammatory changes or DCM; the picture of ARVC (according to TFC-2010 [2]) in combination with LVNC (Figure 3) was found. Thus, the presence of LVNC was verified by cardiac CT and MRI.

Thus, according to the results of the complex examination, the patient had one major (indexed RV volume according to MRI over 100 mL, combined with dyskinesias and decrease of EF up to 25%) and three minor (sustained VT, negative T waves in the left precordial leads and late potentials) criteria of ARVC, signs of LVNC according to cardiac MRI and CT. At the same time, there was an episode of prolonged fever after acute respiratory infections without adequate treatment in the disease onset, decompensation of heart failure provoked by upper respiratory tract infection, angina pain with unchanged coronary arteries, extensive areas of late gadolinium enhancement on MRI (interpreted ambiguously), as well as late contrast agent enhancement on CT (in the middle and

subendocardial layer of the LV) and significant increase in titers of anticardiac antibodies (including specific ANF 1: 160). All these features required exclusion of active myocarditis as a cause of systolic dysfunction, along with two verified cardiomyopathies.

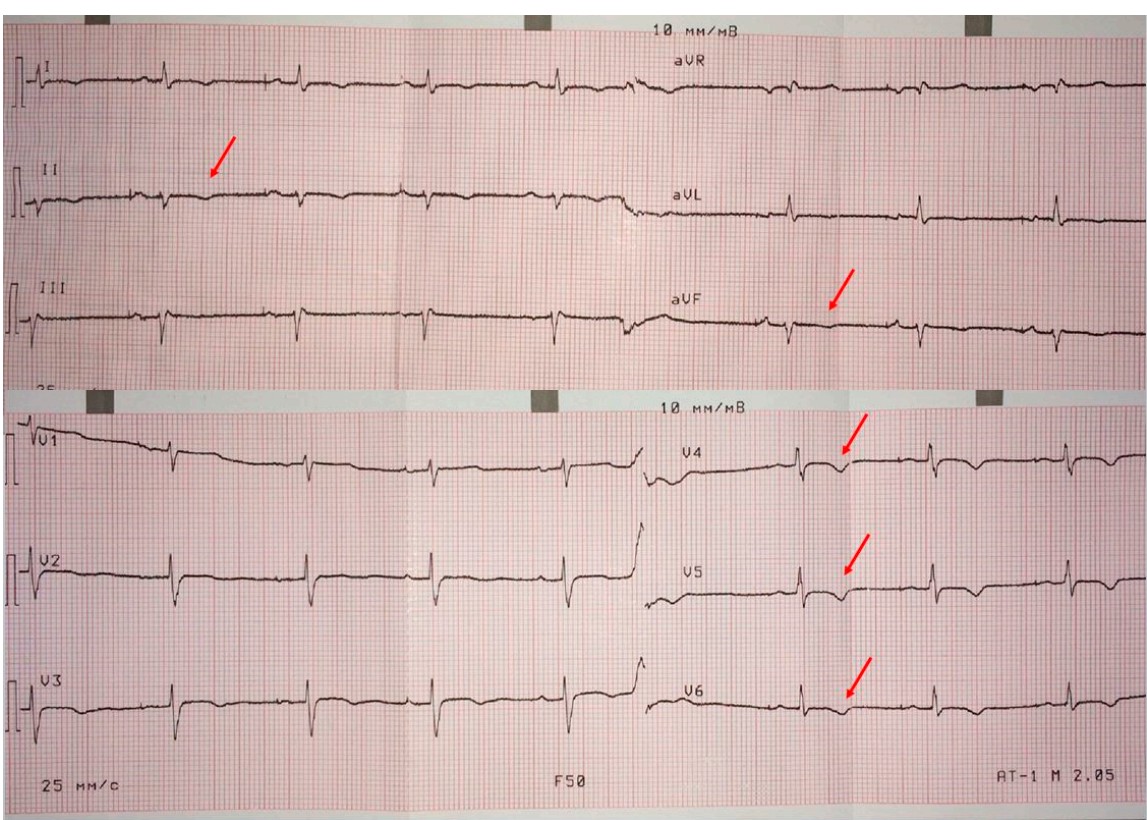

**Figure 2.** Patient's ECG. Low QRS voltages in limb leads, negative T wave in left precordial leads (minor ARVC criterion) and in inferior leads.

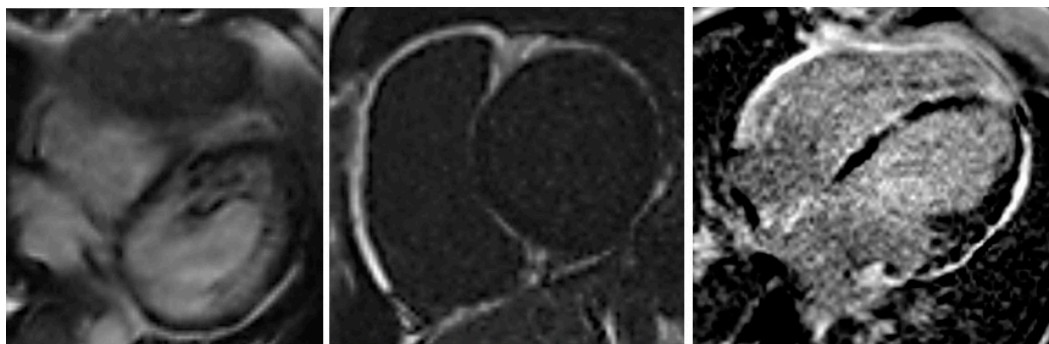

**Figure 3.** Cardiac MRI: left ventricular (LV) end-diastolic diameter (EDD) 66 mm, LV end diastolic volume EDV 243 mL, EF 41%, RV EDD 48 mm, RV EDV 115 mL/m$^2$, EF 25%, areas of hypo/diskinesia of the front wall of RV, in the early and late phases—subepicardial areas of late gadolinium enhancement throughout LV, in IVS from the side of RV and along the walls of RV.

To verify myocarditis endomyocardial, biopsy from the free wall of RV was obtained: endocardium was thin and contained adipocytes in subendocardial area (less than 10%); cardiomyocytes were irregularly hypertrophied, with dystrophic changes in cytoplasm, separated by fibrous septa with focal lymphohistiocytic infiltrates (>14 cells/mm$^2$); Tebizia vessels with edematous endothelium and vasculitis phenomena. No genome of cardiotropic viruses (herpetic group viruses, parvovirus B19 and cytomegalovirus) was found in my-

ocardium. Thus, in addition to the combination of ARVC with LVNC, active chronic infectious-immune myocarditis was confirmed.

### 2.6. Genetic Counseling and Testing

The patient was consulted by a geneticist. DNA-diagnostic using ARVC and sarcomeric panels was recommended. A new genetic variant (NM_004415.4: c.1141-2A>G/N (rs794728111)) was identified in the DSP (desmoplakin) gene. According to bioinformatic analysis, it is a pathogenic splice mutation, which is another major criterion of ARVC.

### 2.7. Follow-Up and Treatment

Antiarrhythmic (sotalol 160 mg/day), cardiotropic (bisoprolol 5 mg/day, perindopril 2.5 mg/day) and diuretic therapy (thorasemide 5 mg/day, spironolactone 50 mg/day) were prescribed, on the basis of which the patient noted significant decrease of dyspnea, regression of leg edema and improvement of general well-being. Due to an episode of tachycardia in the ICD memory, which did not allow unequivocal judgement about presence or absence of atrial fibrillation; the existence of LVNC, which is a risk factor for thrombosis; and a decrease of LV EF to 37%, anticoagulant therapy with rivaroxaban 20 mg/day was prescribed. Because of the patient's family circumstances, immunosuppressive therapy of myocarditis (IST) was not started immediately; spontaneous normalization of Ab titers was observed in March 2014 (Table 1).

**Table 1.** Follow-up of patient I.

| Parameter | December 2013 | March 2014 | April 2015 | January 2016 | March 2018 | November 2019 |
|---|---|---|---|---|---|---|
| ANF | 1:320 | 1:40 | 1:80 | negative | 1:80 | 1:40 |
| AbE | 1:160 | 1:80 | 1:80 | 1:80 | 1:80 | 1:80 |
| AbC | 1:160 | 1:80 | 1:160 | 1:80 | 1:80 | 1:80 |
| AbSM | 1:160 | 1:80 | 1:160 | 1:80 | 1:80 | 1:80 |
| AbCF | 1:320 | 1:80 | 1:160 | 1:80 | 1:80 | 1:80 |
| Dyspnea/leg edema | ++/− | +/+ | +++/++ | +/+− | ++/+− | +/− |
| LV EF | 32% | 35% | 30% | 40% | 46% | 38% |
| ICD | − | − | 2 shocks | − | − | − |
| Antiarrhythmic drugs | sotalol 160 mg/day | | amiodaron 400 mg/day | sotalol 160 mg/day | sotalol 320 mg/day | sotalol 160 mg/day |
| Immunosupressive therapy | no | no | azathioprine 150 mg/day methylprednisolone 4 mg/day | methylprednisolone 8 mg/day | azathioprine 100 mg/day with cancellation due to pneumonia in July 2019 | azathioprine 100 mg/day |

Ab—antibodies; ANF—specific antinuclear factor to cardiomyocyte nuclei (normally negative); AbE—Ab to endothelial antigens (N ≤ 1:40); AbC—Ab to cardiomyocyte antigens (N ≤ 1:40); AbSM—Ab to smooth muscle antigens (N ≤ 1:40); AbCF—Ab to cardiac conductive fibers antigens (N ≤ 1:40); ICD—implantable cardioverter defibrillator; LV EF—left ventricular ejection fraction.

Due to the absence of clinical deterioration, IST was not administered until spring 2015, when dyspnea increased significantly, leg edema appeared again, LV EF decreased to 30% and anticardiac Ab titers increased. Besides, twice appropriate ICD shocks because of VT with heart rate 210/min (Figure 4) with transformation into ventricular fibrillation were registered, and PVBs number according to 24 h ECG monitoring exceeded 4000/day.

The condition was considered as exacerbation of chronic myocarditis. Active IST was started (low doses of methylprednisolone—4 mg/day combined with azathioprine 150 mg/day) and sotalol was temporarily replaced by amiodarone (permanent treatment with amiodarone is impossible, due to photosensitization while taking amiodarone). As a result, a significant clinical and laboratory effect was achieved: dyspnea decreased, leg edema regressed, LV EF stabilized at 40%, VT was completely suppressed, blood tests for anticardiac Ab in January 2016 showed almost no signs of myocarditis immunologic activity.

The patient continues to be monitored annually in FTC and her condition is determined not so much by ARVC progression, but by the activity of superimposed myocarditis.

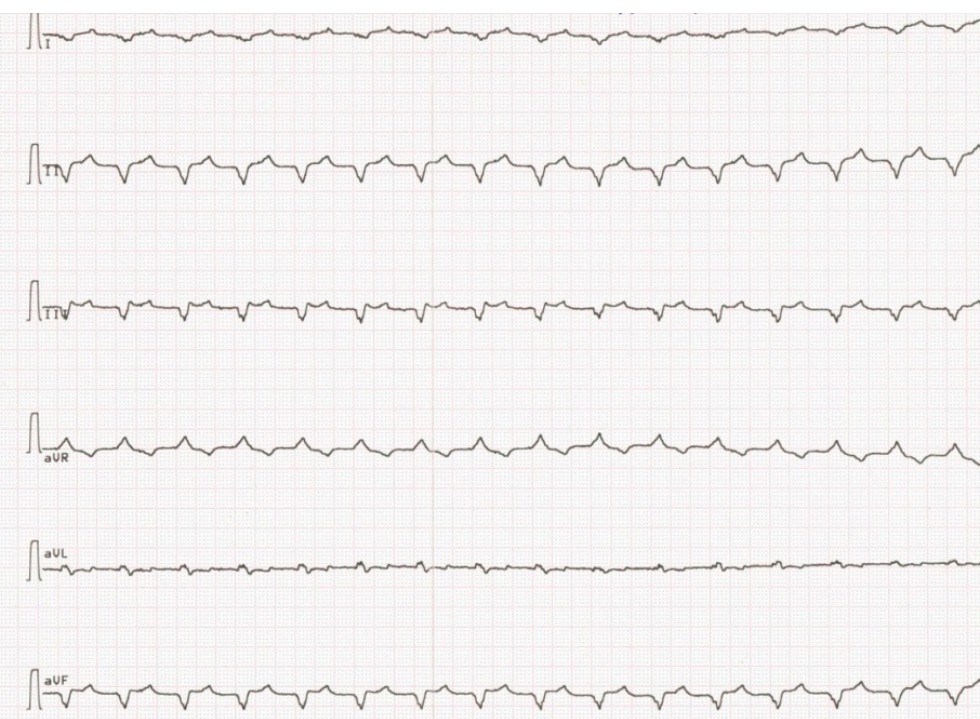

**Figure 4.** Paroxism of sustained ventricular tachycardia.

## 3. Discussion

In review of the literature, we encountered only a few cases of the combination of ARVC and LVNC. In 2006 and 2015, Turkish scientists published two such cases (two young men, in one of them, the presence of LVNC was verified only by EchoCG) [15,16]. One case (a woman who underwent heart transplantation due to the development of refractory heart failure) was described in Italy [17]. A group of Polish scientists in 2009 described nine patients with signs of ARVC in whom a more detailed examination revealed LVNC, which, according to the researchers' opinion, mimicked the clinic of ARVC [18]. This publication has been widely discussed in the scientific community, including by experts such as Finsterer and Stöllberger [19]. In this paper, the criteria for diagnosis of LVNC on MRI and the frequency of confirmation of LVNC by EchoCG and by MRI were not clear. Besides, the presence of LVNC did not explain dilatation of RV in all nine patients, and there were no data on DNA diagnoses of both ARVC and LVNC. It is possible that some patients had a true combination of the two cardiomyopathies, which the authors do not admit. There are data on combinations of ARVC with RVNC, but criteria of "RV noncompaction" have not been developed, so it is considered that increased RV trabecularity is a variant of normal [20,21]. At the same time, none of the above-mentioned publications contains any data on the presence or absence of superimposed myocarditis in patients.

In 2018, we published an article in which the combination of ARVC and LVNC was described in eight patients, and we considered it as a special clinical variant of ARVC [22]. Currently, we have nine such patients in our registry, representing 15.5% of all our patients with ARVC (the ARVC registry includes 58 patients) and 7.2% of patients with LVNC (the LVNC registry contains of 125 patients). Superimposed myocarditis was diagnosed in eight out of nine patients (88.9%), which allows consideration of the combination of these two cardiomyopathies as a favorable background for myocarditis. Patients with a combination of ARVC and LVNC revealed peculiarities of the clinical course of the disease, which distinguished them from the patients with isolated ARVC or LVNC. Life-threatening

ventricular arrhythmias, resistant to antiarrhythmic drugs and leading to appropriate ICD interventions (shocks were recorded in 100% of patients with ICD), were noteworthy. Dilation of RV according to EchoCG, low QRS voltages on ECG, AV block and absence of signs of LV hypertrophy on ECG allowed us to suspect ARVC in patients with verified LVNC. On the opposite hand, in patients with verified ARVC, LV dilatation with decreased LV EF can indicate the presence of LVNC, although it can be observed in patients with biventricular variant of ARVC as well. Nevertheless, the biventricular variant of ARVC by itself does not exclude the presence of LVNC. In 2020, a group of Italian scientists from Padua proposed updated criteria for ARVC that are more sensitive to detect biventricular and dominant-left ARVC variants [23]. According to the criteria they propose, the patient described in this paper has classical signs of biventricular ARVC: morphofunctional and structural changes of both RV and LV.

Another indication for the possible presence of LVNC in patients with ARVC with LV involvement could be the detection of mutations in the *DSP* gene, as changes in this gene were detected in one third of patients with a combination of ARVC and LVNC in our cohort (22.2%—potentially pathogenic variants and 11.1%—variant of uncertain significance). A group of Spanish scientists also described a terminating mutation c.1339C > T in the *DSP* gene detected in three probands and 15 family members of patients with dominant-left ARVC; LVNC was detected in five of them [24].

The prevention of sudden cardiac death (SCD) is particularly important in the management of patients with combination of ARVC and LVNC, since all these patients, despite adequate antiarrhythmic therapy, have a high risk of SCD. Thus, in our patients, there were two appropriate ICD shocks due to sustained VT with transformation into ventricular fibrillation. Diagnostics and treatment of superimposed myocarditis are of great importance, since it aggravates the existing rhythm disorders (sustained VT in the described clinical case was observed only in periods of high activity of myocarditis), as well as contributes to progression of heart failure. There is an opinion that LVNC in patients with myocarditis reflects only secondary rearrangement of myocardial structure due to its inflammatory dysfunction, but the presented observation clearly demonstrates the possibility of combination of genuine myocarditis (verified by EMB and responding to IST) and true LVNC with ARVC caused by mutation in *DSP* gene.

## 4. Conclusions

The combination of ARVC and LVNC is not rare and represents a special form of cardiomyopathy, which is a favorable background for superimposed myocarditis development. Various imaging modalities, in particular, cardiac MRI and CT, play a vital role in the diagnosis. However, verification of myocarditis and determination of indications for its treatment in patients with primary cardiomyopathies is impossible without myocardial biopsy. The primary importance in the management of such patients is the prevention of SCD, since they are at risk of life-threatening ventricular rhythm abnormalities. In addition, prompt diagnosis and treatment of superimposed myocarditis are necessary, since its presence has a significant impact on the clinical course of the disease and prognosis.

**Author Contributions:** Conceptualization (idea, treatment and follow-up of the patient described in the case), Y.L. and O.B.; methodology, O.B.; genetic consulting and testing, A.S. and E.Z.; formal analysis, Y.L. and N.V.; investigation, S.A. (cardiac MRI), N.G. (cardiac CT), E.K. (morphologist who analyzed EMB) and V.S. (EchoCG); data curation, Y.L., O.B. and N.V.; writing—original draft preparation, Y.L.; writing—review and editing, Y.L., O.B. and A.N.; supervision, O.B.; project administration, E.Z. and A.N.; funding acquisition, E.Z. All authors have read and agreed to the published version of the manuscript.

**Funding:** This study (performing DNA diagnostics) was supported by Grant No. 16-15-10421 of the Russian Science Foundation.

**Institutional Review Board Statement:** The study was conducted according to the guidelines of the Declaration of Helsinki, and approved by the Ethics Committee of I.M. Sechenov First Moscow State Medical University (Sechenov University) (protocol code 11-15, date of approval 16 February 2015).

**Informed Consent Statement:** Informed consent was obtained from all subjects involved in the study.

**Data Availability Statement:** The data presented in this study are available on request from the corresponding author. The data are not publicly available because this publication describes a clinical case and the name of the patient should not be disclosed.

**Acknowledgments:** We thank Valentin Sinitsyn for meticulous reanalysis of cardiac MRI disk of the patient.

**Conflicts of Interest:** The authors declare no conflict of interest.

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
