# Peer review of "Three Myocardial Diseases in One Heart: Arrhythmogenic Right Ventricular Cardiomyopathy, Left Ventricular Noncompaction and Myocarditis"

_cardiogenetics, doi:10.3390/cardiogenetics11010003_

Round 1

Reviewer 1 Report

The authors present a very interesting clinical case describing the young patient with signs of arrhythmogenic cardiomyopathy (AC), myocarditis and left ventricular non-compaction. The paper is well written and reflects the current knowledge of those three cardiomyopathies. However, I have a concern: to my opinion the diagnosis of left ventricular non-compaction cardiomyopathy (LVNC) is a little bit speculative in this case. LVNC is a controversial entity. Though, almost three decades has passed since its recognition there are still several open debates including the absence of reliable diagnostic criteria and therefore LVNC is often overdiagnosed. The authors did not find the signs of LVNC based on Echo and initial MRI data. The signs of LVNC were found when the patient underwent cardiac CT and then MRI was revisited in the favor  of LVNC. I would recommend to be more cautious in definition of the findings (trabeculations in left ventricle) as revealed non-compaction might mimic AC with left ventricular involvement and not be the separate heart disease. It seems to me that there are not enough data to verify the LVNC cardiomyopathy in this case.

Several minor comments:

  • Line 42 (Abstract) – abbreviation ARVD, the authors use in the text the abbreviation ARVC.
  • Line 107 – what is the exact size of LV?
  • Of what morphology was the sustained ventricular tachycardia?
  • From which part of the heart the biopsy was taken (as for ARVC diagnosis it is important to have samples from free wall of RV)?
  • The Padua criteria have been proposed and are being discussed, there is no consensus yet among experts. It is not quit right to mention those criteria as “new” yet.

Reviewer 2 Report

Dear Authors,

I have read the manuscript „Three Myocardial Diseases in One Heart: Arrhythmogenic Right Ventricular Cardiomyopathy, Left Ventricular Noncompaction and Myocarditis with great attention". The topic is important, however there are some flaws regarding this submission:

  1. The abstract is too detailed and chaotic and the same time. The Authors should present comprehensively the main outcomes and key conclusion here, not the detailed results, which are duplicated later in the main text. There are some unclear statements, e.g. as the described patient is 34-year- old, the phrase “from 23 years...” means since the patient was 11-year-old? 
  2. Introduction is well-written; I would only suggest to change the term “nosologies” for just “illnesses”, “co-morbidities” or perhaps “nosological entities”
  3. The Case presentation paragraph needs some linguistic corrections - present simple tense is often used, which seems odd. I would also suggest the use of the English term “dyspnea” rather than Greek “dyspnoe”.
  4. In the Follow-up subsection, you report relatively high doses of sotalol given (160 mg per day in the text, however in the Table 1 320 mg per day in March 2018 is shown), with temporary switch to amiodarone on April 2015. Sotalol is generally contraindicated in patients with heart failure with decreased LVEF, and in this case the reported EF varied from 30 to 46%. I presume the deterioration of the LVEF in 2015 was the rationale for the switch to amiodarone, yet the reason for restitution of high-dose (320 mg per day) sotalol is unclear, and the antiarrhythmic medication strategy should be discussed.
  5. I would suggest to change the abbreviation used for sustained VT (“SVT” in the manuscript). Since the “SVT” abbreviation is commonly used for supraventricular tachycardia, this can be misleading, thus the use of “sVT” or “sustVT” instead of “SVT” would be reasonable.

This critical comments do not affect the value of your work, which I appreciate.

They only express the need for general improvement of your manuscript.
